# Effects of Ti Containing Cu-Based Alloy on Sintering Mechanism, Element Diffusion Behavior and Physical Properties of Glass-Ceramic Bond for Cubic Boron Nitride Abrasive Tool Materials

**DOI:** 10.3390/mi14020303

**Published:** 2023-01-24

**Authors:** Xianglong Meng, Bing Xiao, Hengheng Wu

**Affiliations:** 1College of Mechanical and Electrical Engineering, Nanjing University of Aeronautics and Astronautics, Nanjing 210016, China; 2School of Mechanical Engineering, Jiangsu University of Science and Technology, Zhenjiang 212003, China

**Keywords:** sintering behavior, brazing alloy, low temperature co-fired ceramics, elements segregation, element diffusion

## Abstract

Ti containing Cu-based (TC) alloy reinforced glass-ceramic bond was fabricated for cubic boron nitride (CBN) abrasive tool materials, and its crystal composition, phase transformation, sintering activation energy, microstructure, element diffusion mathematical model, physical properties, and the bonding mechanism between the TC alloy reinforced glass-ceramic bond and the CBN grains were systematically investigated. The results showed that the structure, composition and sintering behavior of glass-ceramic were influenced by TC alloy adding. The generated TiO_2_ affected obviously the precipitation of β-quartz solid solution Li_2_Al_2_Si_3_O_10_, thus improving the relative crystallinity, mechanical strength and thermal properties. By establishing the mathematical model for element diffusion, the element diffusion coefficients of Ti and Cu were 7.82 and 6.98 × 10^−11^ cm^2^/s, respectively, which indicated that Ti diffused better than Cu in glass-ceramic. Thus, Ti^4+^ formed a strong Ti–N chemical bond on the CBN surface, which contributed to improving the wettability and bonding strength between CBN and glass-ceramic bond. After adding TC alloy, the physical properties of the composite were optimized. The porosity, bulk density, flexural strength, Rockwell hardness, CTE, and thermal conductivity of the composites were 5.8%, 3.16 g/cm^3^, 175 MPa, 90.5 HRC, 3.74 × 10^−6^ °C^−1^, and 5.84 W/(m·k), respectively.

## 1. Introduction

Cubic boron nitride (CBN) is widely used as an abrasive particle for high speed, high precision and high efficiency grinding due to its high hardness, wear resistance and stable chemical stability [1,2,3]. CBN abrasive tools contain CBN grains and bonding materials which include metals [4], resins [5] and silicate materials [6,7,8]. As a branch of silicate materials, a vitrified bond has the advantages of good self-sharpening, corrosion resistance and low chemical affinity to metals. In order to meet the demands of new workpiece materials and techniques, it is necessary to develop high-performance vitrified bond CBN composites. In the composition of the vitrified bond CBN abrasive tools, the vitrified bond plays the role of shaping, forming holes and holding the abrasives. Therefore, to a certain extent, the performances of CBN abrasive tools rely on the physical properties of the vitrified bond [9].

The vitrified bond is a material with a spatial network structure formed by oxides such as SiO_2_, B_2_O_3_, Al_2_O_3_, etc. as the main body. At present, the vitrified bond systems with different oxides as the main body have been studied, such as Li_2_O–Al_2_O_3_–SiO_2_ [10], CaO–B_2_O_3_–SiO_2_ [11], CaO–Al_2_O_3_–SiO_2_ [12] and MgO–Al_2_O_3_–B_2_O_3_ [13]. These vitrified bond systems have advantages, such as low melting point, high strength and good fluidity, to bind the super hard abrasives such as CBN and diamond. To further improve the physical properties of the vitrified bond, metal oxides have been added to the commonly used systems for modification [14,15,16,17]. An important role of the added phases is to induce precipitation in the matrix, thus obtaining a new phase with properties, such as high flexural strength [6,7,18] and low coefficient of thermal expansion (CTE) [19], which are not present in the matrix. However, the precipitation behavior needs to be screened to exclude those that are against the improvement of the matrix’s physical properties. Shi et al. researched that Al_2_O_3_ content affects the vitrified bond network structure and inhibits the precipitation of Al_2_SiO_5_ and α-SiO_2_, thereby increasing the intrinsic flexural strength of 113 MPa and the CTE of 6.81 × 10^−6^ °C^−1^ [20]. Yu et al. investigated the effects of TiO_2_ and the high magnetic field sintering process on the properties of vitrified bond and showed that TiO_2_ improves the fluidity and flexural strength of the vitrified bond, and the high magnetic field sintering process promotes the densification of the sintered material and inhibits the crystallization of NaAlSi_3_O_8_ [21]. Cui reported that the addition of Y_2_O_3_ increases the glass transition temperature and crystallization temperature, while increasing the flexural strength (202 MPa) and micro-hardness (5.98 GPa) of the vitrified CBN composite [22].

Another way to obtain glass-ceramic material is to make a co-fired material to which a second phase is added in a vitrified bond, which is obtained by the melt-quenching method, thus inducing microcrystal precipitation in the composite and obtaining better physical properties. Luo et al. added Si_3_N_4_ in CaO–Al_2_O_3_–SiO_2_ glass, and the flexural strength and thermal conductivity of the composite reach 122 MPa and 4.22 W/(m·K), respectively [23]. Meng et al. prepared Al_2_O_3_/vitrified bond composites, and the glass transition temperature of the composite and the residual compressive stresses of the generated BaAl_2_Si_2_O_8_ increases with the addition of Al_2_O_3_, when the Al_2_O_3_ content is 57.5 wt%, the flexural strength and the CTE of the composite reach up to 169 MPa and 3.83 × 10^−6^ °C^−1^, respectively [24]. For metallic materials, Cu, Ni, and Zn are also added as a second phase in the vitrified bond, to improve the physical and mechanical properties, such as fluidity and bending strength [25,26,27,28]. Cu–Sn–Ti brazing alloy has excellent high-temperature wettability and bonding strength for diamond and CBN, and is widely used as a metal bond in the preparation of single-layer super abrasive tools [29,30,31]. Therefore, the brazing alloy has good high-temperature wetting behavior on CBN and can form high-strength chemical bonds on the surface of CBN, thereby improving the interfacial bonding strength of CBN composites. Therefore, in this paper, the brazing alloy is used to modify the bonding material. On the one hand, the modification of vitrified bonds by solder alloy can improve the physical properties of composite bond materials. On the other hand, Ti in the solder alloy can form strong chemical bonds with CBN, thereby improving the interfacial bonding characteristics of the composite material. Furthermore, there was no report on the preparation of Cu–Sn–Ti brazing alloy reinforced glass-ceramic bond for CBN abrasive tool materials.

As a super-hard abrasive binder material, good mechanical properties are the fundamental conditions for completing the grinding process. In addition, improving the bonding strength between CBN abrasive and binding material is very important to improve the service life of CBN abrasive. There were two main reasons to add TC alloy to the matrix. First, Cu^2+^ in TC alloy could affect the functional groups of the matrix and thus improve its network structure, resulting in a better intrinsic strength of the matrix. The generated TiO_2_ acted as a nucleating agent that affected the content, structure, and sintering behavior of the TC composites, thus obtaining better physical properties, such as CTE, flexural strength, and thermal conductivity. Second, Ti^4+^ ions had the potential to form strong chemical bonds with CBN, thus improving the bonding strength of the TC/CBN composite. In this study, Ti-containing Cu-based (TC) alloy was employed as a reinforcement phase to fabricate TC-reinforced glass-ceramic bond for CBN abrasive tool materials, in which the glass-ceramic composite acted as the matrix material. The sintering behavior, crystal composition, phase transformation, element diffusion behavior, microstructure, physical properties, and bonding mechanism of TC and TC/CBN composites were systematically investigated.

## 2. Materials and Methods

### 2.1. Sample Preparation

The composition of the TC composite is listed in Table 1. The reagent powders used to fabricate the vitrified bond are SiO_2_, Al_2_O_3_, H_3_BO_3_, BaCO_3_, Na_2_CO_3_, Li_2_CO_3_, ZnO, and MgO (analytical-grade, 0.5 ± 0.05 μm, Xilong Scientific Co., Ltd., Shantou, China). SiO_2_, Al_2_O_3_, and B_2_O_3_ played a skeleton role, and the remaining oxides were used to adjust the viscosity and refractoriness of the basic vitrified bond, so as to match the sintering temperature of the vitrified bond and the brazing alloy. The raw reagent powders were weighted, mixed and dry-type ball milled for 2 h with a ball-to-powder ratio (BPR) of 10:1 to achieve homogeneity. Then the mixed powder was filled into an alumina crucible and heated to 1400 °C for 2 h with a heating rate of 10 °C/min. The molten glass was subsequently quenched in water to form a glass cullet. After drying at 100 °C for 8 h, the glass cullet was ball milled in ethyl alcohol for 8 h with the BPR of 10:1. Afterward, the as-received vitrified bond powder was dried at 100 °C for 2 h and sieved through a 200-mesh sieve. The mixture of the vitrified powder and Al_2_O_3_ (~0.2 μm, Shanghai Chaowei Nanotechnology Co., Ltd., Shanghai, China) with a weight ratio of 1:1.35 were mixed and ball milled in ethyl alcohol for 8 h with the BPR of 10:1. Then, the glass-ceramic, the matrix material, was obtained after drying at 100 °C for 2 h and sifting by a 200-mesh sieve. Ti-containing Cu-based alloy (Changsha Tijo metal materials Co., Ltd., Changsha, China) and the matrix material powder were mixed and ball milled in ethyl alcohol for 1 h with the BPR of 5:1. The microstructure and elemental composition of TC powder are shown in Figure 1. Afterward, TC-reinforced glass-ceramic powder was obtained after drying at 100 °C for 2 h and sieving through a 200-mesh sieve. The specimens were labeled TC0, TC2, TC4, TC6, TC8, and TC16, corresponding to varying amounts of TC (0, 2, 4, 6, 8, and 16 wt%).

The TC powder was then mixed with 5 wt% polyvinyl alcohol (PVA, 1750 ± 50, Sinopharm Chemical Reagent Co., Ltd., Shanghai, China) and pressed at 30 MPa to different dimensions. CBN grits (~0.5 mm, Zhongnan Jiete Super-abrasives Co., Ltd., Zhengzhou, China), TC/glass-ceramic powder, and PVA were mixed with a weight ratio of 30:100:5 and pressed at 30 MPa. The dimensions for flexural strength, thermal conductivity, and CTE measurements were 4 mm × 4 mm × 35 mm, Ø25 mm × 5 mm, and 4 mm × 4 mm × 25 mm, respectively. The sintering temperature was increased from room temperature to the final temperature for 1 h at a heating rate of 10 °C/min. To fully burn off the PVA, a holding period of 20 min was carried out when the sintering temperature reached 350 °C.

### 2.2. Characterization Methods

Crystal composition of the TC and TC/CBN composites was determined by X-ray diffraction (XRD, Bruker Advance D8, Karlsruher Germany). The angular (2*θ*) range of the XRD diffraction patterns was recorded from 10° to 80°. Fourier transform infrared spectrometry (FTIR, Thermo Scientific Nicolet iS20, Waltham, MA, USA) was used in recording the generation of the functional units in the sintered TC composite. X-ray photoelectron spectroscopy (XPS, Axis Ultra DLD, Kyoto, Japan) was employed to verify the formation of functional units and chemical bonds in the sintered samples. All the powders for XRD FTIR and XPS testing are obtained from the sintered bulk materials, which underwent crushing, grinding, and sieving. The exothermic and endothermic behaviors of the TC composite were measured from 30 to 1000 °C with a heating rate of 10 °C/min by differential scanning calorimetry (DSC, Netzsch STA449 F5, Bavarian, Germany). The microstructure and element distribution of the TC and TC/CBN composites were observed through a scanning electron microscope (SEM, Hitachi Regulus 8100, Tokyo, Japan) in conjunction with energy dispersive spectrometry (EDS, Bruker XFlash 6-60, Karlsruher, Germany). The bulk density and porosity of the sintered TC composite were evaluated using the Archimedes method, and calculated based on Equations (1) and (2), respectively.
(1)$$\rho =\frac{m}{V}$$
(2)$$\chi =\frac{V_{0}-V}{V}\times 100\%$$
where *ρ* is the bulk density (g/cm^3^), m is the weight of the tested sample (g), *V* is the volume of the sintered testing sample (cm^3^). For Equation (2), *χ* is the porosity of the tested sample (%), *V*_0_ is the volume of the un-sintered sample (cm^3^), and *V* is the volume of the sintered sample. Three-point flexural experiments were performed with a universal testing machine (Suns UTM 4502X, Shenzhen, China) at a cross-head speed of 0.5 mm/min. Flexural strength *σ* was calculated based on Equation (3):(3)$$\sigma =\frac{3PL}{2bh^{2}}$$
where *P* is the break force (N), *L* is the span (mm), and *b* and *h* are the width and thickness of the specimens (mm), respectively. The Rockwell hardness was performed by a hardness tester (THR-150 M, China) with a load of 150 kgf maintained for 15 s. The thermal expansion behavior and the CTE (30–300 °C) of the TC composite were recorded by thermal dilatometer (DIL, Netzsch DIL 402 C, Germany) from 30 to 600 °C with a heating rate of 10 °C/min. The thermal conductivity of the TC composite at 500 °C was determined by the transient plane heat source method (Hot Disk TPS 2500 S, Uppsala, Sweden).

## 3. Results and Discussions

### 3.1. Sintering Mechanism of TC Composites

#### 3.1.1. XRD

Figure 2a illustrates the XRD patterns of the TC composites with different TC contents sintered at 950 °C for 1 h. The crystallization phase of the TC composites consisted of α-Al_2_O_3_ (PDF#: 74-1081), Li_2_Al_2_Si_3_O_10_ (PDF#: 73-2335), BaAl_2_Si_2_O_8_ (PDF#: 74-0177), TiO_2_ (PDF#: 87-0710), and Sn_11_Cu_39_ (PDF#: 71-0122). Figure 2b illustrates the XRD patterns of the TC composites with 16 wt% TC content sintered from 750 to 1000 °C for 1 h. Similar to the results in Figure 2a, Li_2_Al_2_Si_3_O_10_ appeared in samples at all sintering temperatures, except for LiAlSi_2_O_6_ (PDF#:73-2336) which appeared in the sample sintered at 1000 °C. When the sintering temperature was 1000 °C, the viscosity of the glass-ceramic decreased, the material migrated sufficiently and the precipitation of SiO_2_ increased, so that LiAlSi_2_O_6_ (Li_2_O·Al_2_O_3_·4SiO_2_, high quart solid solute) was therefore produced instead of Li_2_Al_2_Si_3_O_10_ (Li_2_O·Al_2_O_3_·3SiO_2_). Combining with Figure 2a, it can be seen that the diffraction intensity of TiO_2_ (2*θ* = 27.44º) sintered at 800 °C was much higher than that of TiO_2_ sintered at 750 °C, and increased with TC content, which indicated that TiO_2_ formed at about 800 °C. The diffraction peak intensity of Li_2_Al_2_Si_3_O_10_ at 2*θ* = 25.51º sintered at 800 °C was therefore much higher than that sintered at 750 °C, due to TiO_2_ acting as a nucleating agent to promote the crystallization of Li_2_Al_2_Si_3_O_10_.

Furthermore, the detailed crystal phase composition and relative crystallinity of the samples with the TC alloy varied from 2 to 16 wt% are presented in Table 2. It can be seen that the relative crystallinity of the samples increased from 68.2 to 75.5%, and the amount of Li_2_Al_2_Si_3_O_10_ increased from 14.4 to 26.4% and that of BaAl_2_Si_2_O_8_ decreased from 48.8 to 32.8%. In the meantime, *α*-Al_2_O_3_ content decreased from 35.3 to 25.9% due to *α*-Al_2_O_3_ being the raw material for the synthesis of Li_2_Al_2_Si_3_O_10_ and BaAl_2_Si_2_O_8_, and the chemical reaction formulae are listed as follows:Li_2_O + Al_2_O_3_ + SiO_2_ → Li_2_Al_2_Si_3_O_10_(4)
BaO + Al_2_O_3_ +SiO_2_ → BaAl_2_Si_2_O_8_(5)
which was explained from the perspective of material composition. In detail, the formation process of Li_2_Al_2_Si_3_O_10_ can be illustrated as follows. The β-quartz crystal is composed of a three-dimensional network of [SiO_4_] tetrahedrons with a hexagonal spiral structure and is widely present in vitrified bonds. The [SiO_4_] tetrahedron structure with regular replacement of Si^4+^ ion by Al^3+^ ion, β-quartz solid solution (Li_2_O·Al_2_O_3_·SiO_2_) is therefore formed. Furthermore, the Al^3+^ ion is located at the lattice site of the original Si^4+^ ion, and Li^+^ ions fill the voids near the Al^3+^ ion to balance the electrical neutrality of the structure [8]. When half of the Si^4+^ ions in β-quartz are replaced, β-eucryptite (Li_2_O·Al_2_O_3_·2SiO_2_) is formed. Since both β-quartz and β-eucryptite have the same hexagonal spiral structure, a continuous solid solution, Li_2_Al_2_Si_3_O_10_ (Li_2_O·Al_2_O_3_·3SiO_2_), can be formed [32]. In previous studies, TiO_2_ played a fundamental role in controlling the crystallization of β-quartz solid solution [33,34]. Kniess et al. prepared Li_2_Al_2_Si_3_O_10_ with the addition of TiO_2_ as the nucleating agent to the base glass of the SiO_2_–Al_2_O_3_–Li_2_O system [35]. Therefore, it is suggested that TC alloy could promote the relative crystallinity of the composites. Moreover, the added TC alloy decreased the BaAl_2_Si_2_O_8_ content but increased that of Li_2_Al_2_Si_3_O_10_, and the crystallization efficiency of Li_2_Al_2_Si_3_O_10_ was influenced by the formed TiO_2_ as a nucleating agent in TC alloy.

#### 3.1.2. DSC and XPS

In order to verify the physical and chemical changes during sintering, the DSC curves of TC composites with different TC alloy contents are measured at a heating rate of 10 °C/min from 30 to 1000 °C, as shown in Figure 3. The endothermic peak 1 at about 630 °C referred to the liquefaction process of the matrix, which was a sign of liquid phase sintering. The exothermic peak 1 at the range of 680–750 °C corresponded to the precipitation of BaAl_2_Si_2_O_8_ and Li_2_Al_2_Si_3_O_10_ and showed an increase in the relative intensity with the increasing of TC content, due to the increased TiO_2_ formation, which promoted the precipitation of Li_2_Al_2_Si_3_O_10._ The exothermic peak 2 at about 785 °C became more apparent with the increasing TC content and corresponded to the formation of TiO_2_. The endothermic peak 2 at about 950 °C was believed to be correlated to the melting process of TC powder. The endothermic process at about 490–530 °C corresponded to the glass transition process. The midpoint of the DSC curve shifting to an endothermic direction was selected as the glass transition temperature, as shown in the insert figure (i). The *Tg* value increased from 511 to 518 °C and then decreased to 510 °C with the increase in TC content (insert figure (ii)). This is consistent with the research of Węgrzyk et.al [25]. As the basic constituent elements of vitrified bond, boron has two functional units in vitrified bond, [BO_3_] triangle units and [BO_4_] tetrahedrons, which affect the network structure, sintering behavior, and physical properties of the vitrified bond. XPS is therefore conducted to verify changes in the functional units of boron in TC0/ and TC4/CBN composites.

Figure 4 shows the XPS spectrum of B 1s for the TC0/ and TC4/CBN composites sintered at 950 °C for 1 h. The characteristic peaks of [BO_4_], [BO_3_], and the N–B bond were observed at 192.6, 191.9 [36], and 190.3 [37] eV, respectively. The N–B bond refers to CBN. Comparing Figure 4a,b demonstrated that the proportion of [BO_3_] triangle units decreases with the increase of [BO_4_] tetrahedrons, which can be explained by the fact that Cu^2+^ ions in the TC affected the coordination number of boron and promoted the conversion from [BO_3_] to [BO_4_] groups by entering the vitrified bond network. At the same time, as an intermediate cation, Ti^4+^ ions enter the vitrified bond network, connect the non-bridged oxygen to build [TiO_4_] tetrahedrons, increase the connection of the vitrified bond network structure, break Si–O–Si and Si–O–Al connections, partly replace Al^3+^ and Si^4+^ from [AlO_4_] and [SiO_4_] [7], resulting in a more compact network structure. All of these contributed to an increase in the network dimensionality and connectivity, resulting in a more compact and rigid glass network, which could explain *Tg* values increased with TC content increased from 2 to 8 wt%. However, when the TC content reached 16 wt%, the excess Cu^2+^ and Ti^4+^ entered the vitrified bond network structure, as a network modifier leading to a loose network structure and reducing the *Tg* values. Therefore, it is suggested that TC alloy promoted the conversion of [BO_3_] to [BO_4_] units, thus affecting the structure of the vitrified bond and the glass transition process of the TC composites.

#### 3.1.3. Calculation of Sintering Activation Energy

The activation energy corresponded to the energy barrier of glass to crystal transition, which was an important index to evaluate the crystallization ability of glass materials. Figure 5 shows the shrinkage curves for TC composite with various TC alloy addition recorded from room temperature to 900 °C using different sintering rates (*α*) of 10, 15, and 20 K/min. The specific linear shrinkages (dL/L_0_) of 2, 4, and 6%, and their corresponding temperatures were recorded to calculate the sintering activation energy of the TC composites with different sintering rates. Furthermore, the activation energy can be calculated by the Arrhenius equation as follows.
(6)$$\ln\alpha =-\frac{E_{a}}{RT}+A$$
where *E_a_* is the sintering activation energy, kJ/mol; R denotes the ideal gas constant, 8.314 J/(mol·K); *T* is the thermodynamic temperature, K; A denotes the Arrhenius parameters. Figure 6 depicts the fitted results of ln *α*~ 1000/*T* with different shrinkages and the calculated sintering activation energy of the TC composites. It illustrates that *E_a_* increased from 124.5–212.9 kJ/mol with TC increased from 2 to 8 wt%. This is mainly because TC alloy affected the structure of the vitrified bond and resulting in a more rigid and compact vitrified bond network structure, which improved the viscosity of the vitrified bond, thus reducing the material migration and wetting behavior, which was not conducive to achieving liquid phase sintering. In addition, it would reduce the precipitation tendency of the vitrified bond with a regular functional group composition and complete network structure. Thus, the *E_a_* values were improved. When TC content was 16 wt%, the excess Cu^2+^ and Ti^4+^ ions entered the vitrified bond network, resulting in a loose structure, thus reducing the viscosity of the vitrified bond and contributing to the material migration. The precipitation tendency of the vitrified bond was improved, which led to a reduced *E_a_* value. It is suggested that a proper amount of TC would be beneficial to the precipitation in vitrified bonds at a relatively low sintering temperature.

### 3.2. Microstructure and Element Diffusion Behavior of TC Composites

#### 3.2.1. Morphology and Porosification Behavior of TC Composites

Figure 7 shows the microstructure of the TC composites with different TC alloy contents. It can be seen that the spherical and ellipsoidal TC powders were uniformly distributed in the matrix. In TC2 and TC4 composites (Figure 7a,b), there were no obvious gaps around the TC powder. In the cases of composites with 6–16 wt% TC contents (Figure 7c–e), there were clear gaps between the TC powder and the matrix. This was because when the sintering temperature exceeded the glass transition temperature, the vitrified bond powder gradually liquefied and wetted the ceramic filler (Al_2_O_3_) and the TC powder along the gaps between the powders in the samples by capillary action. As the gaps between Al_2_O_3_ and the smaller TC particles provided sufficient surface tension, the Al_2_O_3_ and the smaller TC particles were therefore fully wetted by the liquid-state vitrified bond, resulting in a dense interface. However, as the TC content increased, the number of larger TC particles also increased. The gaps between the TC alloy and the Al_2_O_3_ particles became larger, decreasing the surface tension that was provided by the gaps, resulting in insufficient wetting for the larger TC particles. As a result, the gaps appeared at the edges of the larger TC particles.

In addition, the gaps between TC and Al_2_O_3_ particles became wider by the TC particles with large area breakage, causing a lower surface tension for the liquefied vitrified bond, which also resulted in voids and connected pores around the TC particles (Figure 7f,g). Figure 7h shows the surface morphology of the TC16 composite. A crescent-shaped gap was formed between the TC powder and the matrix, which was caused by the different shrinkage of the TC ally and the matrix during cooling, and was another way for pore (void) formation. As shown in Figure 7i, there was a gully-like defect on the surface of the TC particle (showed by the yellow line), and the small area breakage had little effect on the liquid-state vitrified bond surface tension. Thus, the dense interface was still formed between the matrix and the TC particles. These show that the addition of TC alloy had significant effects on the morphology and the porosification of the TC composites.

#### 3.2.2. Establishing of the Element Diffusion Mathematical Model and Calculation of the Element Diffusion Coefficient of TC Composites

Figure 8 shows the microstructure and EDS result of the interface between the TC particle and the matrix. In the EDS results, the scanning area was divided into five parts and marked as A, B, C, D, and E, respectively. Areas A and D were the matrix area. Area B and D were the areas where the matrix covered the TC particle, and Area C was the area where the TC particle was exposed after the fracture. In Areas B and D, the element intensity of Al and Si decreased and that of Cu and Sn increased as the scanning point approached the TC particle, which was the classical element distribution rule for composite. Differently, the element intensity of Ti decreased with the scanning point approaching the TC particle, and its peak intensity was present in the matrix, which covered the TC particle in Areas B and D. In the meantime, the element distributions of Cu and Sn displayed an opposite trend to that of Ti. Comparing Ti distributions in Areas B and C, it could be found that the element intensity of Ti was much higher than that of Cu and Sn. It is suggested that the element segregation of Ti occurred during sintering. Ti was enriched in the matrix near the outer layer of the TC particle (Areas B and D).

For TC composites, the vitrified bond begins to soften and wet the alloy surface as the sintering temperature exceeds the glass transition temperature. The increasing temperature helps to increase the atomic activity. Thus, material diffusion will occur at the interface between the alloy and the liquefied vitrified bond. As a result, the Ti^4+^ and Cu^2+^ ions at the interface will diffuse into the vitrified bond due to their concentration differences. The diffusion process follows Fick’s law, as shown as follows.
(7)$$J=-D\frac{\partial C}{\partial X}$$
(8)$$\frac{\partial C}{\partial t}=\frac{\partial }{\partial X}\left(D\frac{\partial C}{\partial X}\right)$$
where, *J* is the diffusion flux, which is the flux of diffusible material per unit time through a unit cross-sectional area perpendicular to the direction of diffusion. *D* is the diffusion coefficient. *C* is the diffusible material relative content. *X* is the diffusion distance. The two initial conditions for diffusion were defined as follows:(9)$$t=0:\;C{|}_{x=0}=\infty ,\;C{|}_{x\neq 0}=0$$
(10)$$t\neq 0:\;C{|}_{x=\infty }=0$$

Therefore, the diffusible material relative content *C* could be solved as follows:(11)$$C=\frac{a}{\sqrt{t}}\exp\left[\frac{-x^{2}}{4Dt}\right]$$
where *a* denotes a constant.

Figure 9 depicts the data points of Ti and Cu from line scanning EDS analysis and their fitted curves. The starting point for Ti diffusion was chosen at the peak position of signal intensity of Ti^4+^ in the matrix rather than at the alloy interface due to the presence of element segregation of Ti. At the same time, Cu^2+^ ion content at the starting point was the lowest value. Therefore, the Ti^4+^ ion relative content and its position in the matrix followed the fitted formula, as shown as follow.
(12)$$y=-1270.56\;\exp\left[\frac{-x^{2}}{1.1262}\right]+78.63$$
thus, 4*Dt* = 1.1262 μm^2^, where *t* = 3600 s referred to the holding time during sintering. Therefore, *D*_Ti_ = 7.82 × 10^−11^ cm^2^/s, which means the diffusion coefficient of the Ti^4+^ ions in the matrix is 7.82 × 10^−11^ cm^2^/s at 950 °C. Using the same method, the mathematical model of Cu^2+^ ions of relative content and the positions in the matrix was obtained as follows:(13)$$y=642.54\;\exp\left[\frac{-x^{2}}{1.0044}\right]+656.10$$
thus, *D*_Cu_ = 6.98 × 10^−11^ cm^2^/s. The diffusion coefficient of Ti in the matrix was therefore greater than that of Cu in the matrix. It could be explained as follows: According to the Stokes–Einstein relation (14), the diffusion constant *D_c_* of a particle is related to the absolute temperature of diffusion *T*, the viscosity of the medium *η*, and the radius of the diffusing particle *r*.
(14)$$D_{c}=\frac{K_{B}T}{6\pi r\eta }$$
where *K_B_* is Boltzmann’s constant. Due to Ti having a smaller ionic radius than Cu (r_Ti_ = 0.605 Å and r_Cu_ = 0.71 Å), the diffusion constant of Ti^4+^ is, therefore, higher than Cu^2+^, and easier to diffuse in the matrix.

Figure 10 shows the morphology and EDS result of the crater left by the TC particle. The yellow dashed line was the outline of the original alloy particle, and the blue-grey area referred to the interface between the original alloy particle and the matrix. From the EDS result for point B, the elemental mass ratio of Ti to Cu (W_Ti/Cu_% = 22.64%) was higher than that of the un-sintered alloy particle (W_Ti/Cu_% = 15.29%, in Figure 1). In summary, the diffusion coefficient of Ti and Cu were 7.82 × 10^−11^ and 6.98 × 10^−11^ cm^2^/s (950 °C), respectively. Although the actual starting content of Ti was lower than that of Cu, the diffusion of Ti is still greater than that of Cu during interfacial diffusion, which was consistent with the calculation of the diffusion coefficient, which indicated that Ti diffused better than Cu in the matrix.

### 3.3. Physical Properties

#### 3.3.1. Porosity and Bulk Density

Figure 11a shows the porosity and bulk density of TC composites with respect to TC content. For the porosity, when the TC content was less than 4 wt%, the porosity of the TC composite decreased from 6.1% to 5.8%. It was mainly due to the TC content being low and the liquefied vitrified bond being sufficient to wet the TC particles. At the same time, the TC particles filled the pores in the matrix, thus reducing the porosity of the TC composite. In the cases with 6–16 wt% TC content, as analyzed in Section 3.2.1, the proportion of larger TC particles and the particles with large breakage increased, making the gaps in the sample larger and reducing the surface tension of the liquid-state vitrified bond. It resulted in a portion of the TC particles being accompanied around by pores, thus sharply increasing the porosity of the composite to 10.5%. Correspondingly, the density curve tended to be opposite to the porosity curve. The density reaches a maximum of 3.16 g/cm^3^ at a TC content of 4 wt%. It is suggested that large amounts of TC contents were not conducive to reducing the porosity and improving the bulk density of the TC composites.

#### 3.3.2. Flexural Strength and Rockwell Hardness

Figure 11b shows the flexural strength and Rockwell hardness of TC composites with different TC contents. The flexural strength was mainly used to examine the strength of brittle materials and represented the ability of the material to resist bending without fracture. The flexural strength of TC0, TC2, and TC4 composites gradually increased from 169 MPa to 175 MPa. This was due to the addition of TC material obtained from the vitrified bond with a more compact and rigid network structure, which contributed to improving the intrinsic density and strength of the vitrified bond. Moreover, the TC particles inhibited the flow of the liquid-state vitrified bond during sintering and therefore increased the viscosity and the internal density of the liquefied vitrified bond, which led to a higher flexural strength. The generation and propagation of micro-crack were the basis for the formation of macro-cracks. In terms of inhibiting micro-crack generation and propagation, on the one hand, the generated crystals during sintering, such as Li_2_Al_2_Si_3_O_10_, BaAl_2_Si_2_O_8_, TiO_2_, etc. in the TC composite can nail the crack and stop it from continuing to propagate. On the other hand, the crystals in the TC composite can increase the crack propagation distance and dissipate more crack extension energy. These all contributed to the material achieving a higher flexural strength. On the contrary, when the excess crystals were precipitated and the crystals were large in size, the interface between the crystals and the composite gradually became complicated and the composite structure was dissociated, thus weakening the mechanical properties of the composite containing 6–16 wt% TC alloy.

In terms of hardness, the Rockwell hardness curve followed the same trend as the flexural strength curve. The Rockwell hardness of the composite increased slowly when the TC content was less than 4 wt%. For TC4, the hardness reached a maximum value of 90.5 HRC. For the cases containing 6–16 wt% TC alloy, the hardness dropped sharply to 70.2 HRC. The hardness referred to the ability of a material to resist depression and deformation caused by a hard object indentation and was related to the bulk density and porosity of the material. The addition of small doses of TC alloy increased the density of the TC composites, thus increasing the hardness of the TC composites. Moreover, the TC alloy consisted of copper, tin, and titanium, which was much softer than the vitrified bond. Therefore, when the addition of TC alloy excessed 6 wt%, the Rockwell hardness decreased dramatically.

#### 3.3.3. Thermal Conductivity and CTE

Figure 11c shows the thermal conductivity of the TC composites with various TC contents. It showed that the thermal conductivity of the TC composites increased with the addition of the TC alloy from 4.17 W/(m·k). The thermal conductivity of 5.84 W/(m·k) was obtained when the TC content was 6 wt%, which was the maximum growth rate of 16%. According to gas molecule collision theory, the thermal conductivity of crystalline materials is the result of phonon collisions and the specific thermal conductivity *λ* can be calculated by the following equation:*λ* = 1/3*Cv*_*p*_*l*_*p*_(15)
where *C* is the volumetric heat capacity of phonons, *v_p_* is the average velocity of phonons and *l_p_* is the mean free path of phonons. The vitrified bond, as a glass material, has the structural characteristics of short-range order and long-range disorder and can be regarded as a “crystal” consisting of grains with very small lattice spacing. The average phonon-free range of a vitrified bond is, therefore, in most cases, much smaller than that of a crystal. Furthermore, as seen from the results in Table 2, the relative crystallinity of the matrix increased with the addition of the TC alloy and therefore the thermal conductivity of the vitrified bond increased accordingly. The thermal conductivity of TC composites increased continually as the thermal conductivity of TC alloys is much higher than that of the matrix material. However, when the TC alloy was greater than 6 wt%, the TC alloy only reduced the growth rate of the thermal conductivity because the TC alloy increased the porosity of the material due to the very small volumetric heat capacity of air.

The CTE between 30 and 300 °C of the TC composite with various TC content is shown in Figure 11d. The TC composite reached the minimum value of the CTE (*a* = 3.74 × 10^−6^ °C^−1^) when the TC content was 2 wt%. Then, the CTE of the TC composites increased as the TC content increased, indicating that the addition of large amounts of TC alloy was not conducive to obtaining TC composites with low CTE. For glass-ceramic composites, its CTE was determined by the CTE of the crystals, ceramic filler, and residual glass phase produced during sintering. The CTE of the TC composite could be calculated as follows [7]:(16)$$a=\overset{n}{\underset{i=1}{\sum }}a_{1}v_{1}+a_{2}v_{2}+\cdots +a_{i}v_{i}+\cdots +a_{n}v_{n}$$
where *a_i_* is the CTE and *v_i_* is the volume fraction of *i* phase.

When the TC content was 2 wt%, the minimum CTE (*a* = 3.74 × 10^−6^ °C^−1^) of TC composites was obtained. It was due to the fact that Cu^2+^ ions enter the vitrified bond network, promoting the transition from [BO_3_] to [BO_4_] units, while Ti^4+^ ions connect with unsaturated oxygen and form oxygen bridges, increasing the connectivity and rigidity of the vitrified bond network structure. Therefore it was beneficial to reduce the CTE of the residual vitrified bond. In terms of crystals, combining the results of Table 2, the generated Li_2_Al_2_Si_3_O_10_ (*a* = –2.34 × 10^−6^ °C^−1^) and BaAl_2_Si_2_O_8_ (*a* = 2.29 × 10^−6^ °C^−1^) [38] with ultra-low CTE occupied more than 60% mass fraction in the TC composites, which helped to reduce the CTE of the TC composites. However, as the TC alloy content increased, TiO_2_ (*a* = 8.9 × 10^−6^ °C^−1^) [14] and Sn_11_Cu_39_ (*a* = 18.4 × 10^−6^ °C^−1^, calculated from the CTE of Cu and Sn [39]) content also started to increase, so the CTE of the composite was increased dramatically.

### 3.4. Bonding Mechanism between CBN and TC Composite

#### 3.4.1. Microstructure

Figure 12 shows the cross-sectional microstructure of the TC4/CBN composite after flexural strength testing. Figure 12a illustrates that the matrix material was homogeneous in texture with spherical TC powder distributed in it. The presence of a cleavage plane on the CBN section indicated that the CBN particles were locally fragmented under load, providing new edges for the CBN composite abrasive and contributing to improving the self-sharpening of the abrasive tools. Figure 12b demonstrates the enlarged area of the dashed box in Figure 12a, which showed that the CBN grain was completely fused with the TC composite, with no defects such as cracks and small pores, which were conducive to improving the interfacial bonding strength of the TC/CBN composite.

#### 3.4.2. Flexural Strength and XPS

Figure 13 shows the flexural strength of TC/CBN composites with various TC contents. The flexural strength of TC/CBN composites first increased and then decreased with the increasing TC addition and reached the maximum value (92 MPa) when the TC amount was 2 wt.%. The value was about 14% higher than that of TC0.

Figure 14 shows the XPS patterns of the TC4 and TC4/CBN composites sintered at 950 °C for 1h. Figure 14a shows the XPS survey spectra of the TC4 and TC4/CBN composites, with variations at the Ti and N shown by the dashed boxes. Figure 14b shows the XPS spectrum of N 1 s for the TC4/CBN composite. The peaks at 398.3 [37] and 397.6 eV [40] refer to N–B (CBN) and N–Ti bonds, respectively, indicating that TiN was formed on the interface of the TC/CBN composite. Figure 14c shows the XPS spectrum of Ti 2p for TC0/CBN composite, the peaks at 464.4 and 458.4 eV corresponded to Ti^4+^ in TiO_2_, and the peaks at 463.4 and 457.7eV represented Ti^3+^ in Ti_2_O_3_ [14]. Combining with Figure 2a,b, the presence of Ti in the TC composite was in the form of TiO_2_ and a small amount of Ti_2_O_3_. Figure 14d shows the XPS pattern of Ti 2p for the TC4/CBN composite. In addition to the characteristic peaks of Ti+ and Ti^4+^, the presence of characteristic peaks of the Ti–N bond at 460.4 and 455.2 eV [41] was found, corroborating the results of TiN formation at the interface of the TC/CBN composite (Figure 14b). Theoretically, the CBN surface is enriched with B^3+^, N^3–^ and Ti^4+^ ions and, therefore, TiN and TiB_2_ may be generated at the interface, but since the Gibbs free energy for TiN generation is less than that for TiB_2_, only TiN was generated at the interface between the matrix and the CBN grains, contributing to the wettability between the CBN grains and the TC composites.

Theoretically, fractures in composites occur in the weaker parts of the matrix, the second phase, and their interface. In the TC composite, the better intrinsic flexural strength of the TC bond and good interfacial fusion state contribute to the flexural strength of the CBN composite. Although TC4 achieved the highest flexural strength (Figure 11b) and good interfacial integration with CBN (Figure 12), the flexural strength of the TC4/CBN composite was 86 MPa, which is lower than the value of the TC2/CBN composite. It was due to the mismatch in the CTE between the TC composite and the CBN during cooling, resulting in the formation of residual thermal stresses at their interface. The difference in flexural strength between TC2 and TC4 was small, while TC2 (*a* = 3.74 × 10^−6^ °C^−1^) has a CTE similar to that of CBN (*a* = 3.5 × 10^−6^ °C^−1^). The residual thermal stress at the interface of TC2/CBN was therefore less than that of TC4/CBN, and thus the maximum flexural strength of the CBN composite was reached when the TC content was 2 wt%. The formation of TiN at the interface of the TC/CBN composite also promotes the flexural strength of the TC/CBN composite.

## 4. Conclusions

Ti-containing Cu-based alloy was added to a glass-ceramics composite to fabricate CBN abrasive tool materials. The glass-ceramics composite consisted of vitrified bond and ceramic fill (Al_2_O_3_). The sintering kinetics, crystal composition, phase transformation, element diffusion behavior, microstructure, physical properties, and the bonding mechanism of the Ti-containing Cu-based alloy-reinforced glass-ceramics bond CBN composite were systematically investigated. The generated TiO_2_ acted as a nucleating agent, affected and promoted the precipitation of Li_2_Al_2_Si_3_O_10_, and improved the relative crystallinity from 68.2 to 75.5%. Ti^4+^ is enriched in the glass-ceramic bond near the outer layer of the alloy particle. According to the established mathematics model for element diffusion, the element diffusion coefficient of Ti and Cu were 7.82 × 10^−11^ and 6.98 × 10^−11^ cm^2^/s (950 °C), respectively, which indicated that Ti diffused better than Cu in the matrix at 950 °C. The physical properties of the composite were optimized by TC addition. The porosity, bulk density, flexural strength, Rockwell hardness, CTE, and thermal conductivity of the composites were 5.8%, 3.16 g/cm^3^, 175 MPa, 90.5HRC, 3.74 × 10^−6^ °C^−1^, and 5.84 W/(m·k), respectively. A Ti–N bond was formed on the surface of the CBN grain, improving the wettability between the matrix and CBN grains, and the matched CTE of the matrix (3.74 × 10^−6^ °C^−1^) with CBN also contributing to improving the flexural strength of the CBN composite to 92 MPa, which was 14% higher than that of the composite without alloy addition. After the addition of TC alloy, the structure of the matrix material is more compact, and the physical properties of the composite material are further improved, indicating that the TC composite material has the potential to become a high-performance bonding material.

## Figures and Tables

**Figure 1 micromachines-14-00303-f001:**
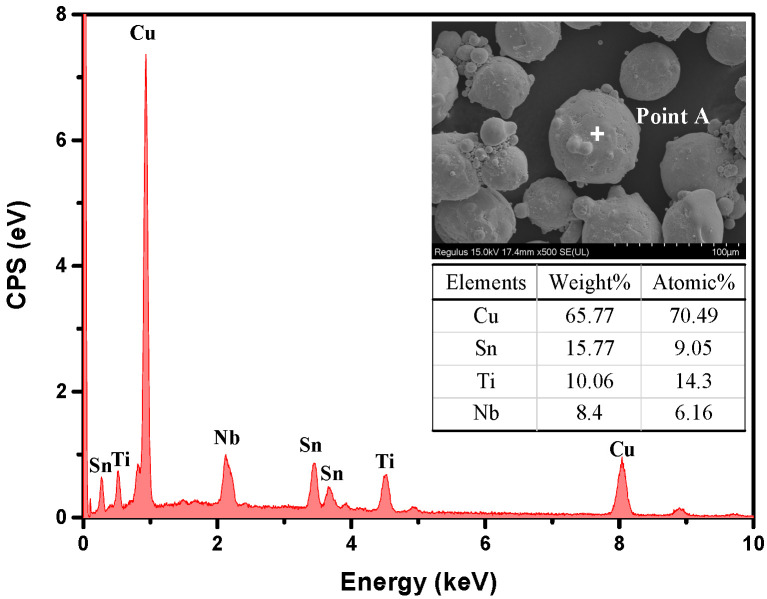
Microstructure and elements composition of TC powder.

**Figure 2 micromachines-14-00303-f002:**
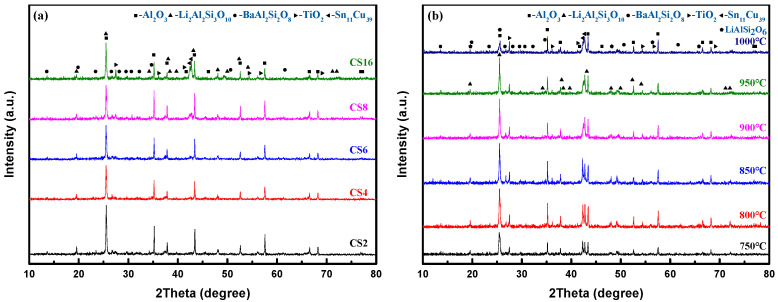
XRD patterns of TC composites: (**a**) the samples containing 2–16 wt% TC sintered at 950 °C; (**b**) the samples containing 16 wt% TC sintered at 750–1000 °C.

**Figure 3 micromachines-14-00303-f003:**
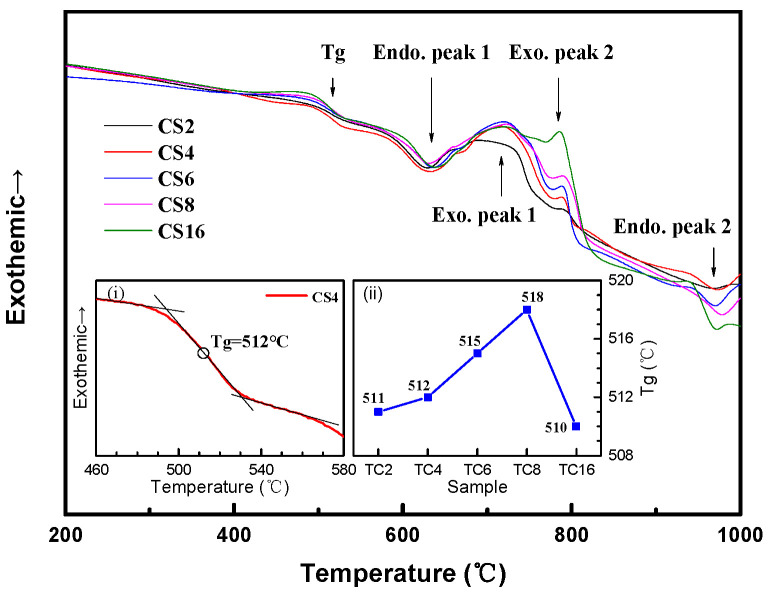
DSC curves of the TC composites with different TC contents at a heating rate of 10 °C/min (i) glass transition temperature of CS 4, (ii) Tg values of the TC composites with different TC contents.

**Figure 4 micromachines-14-00303-f004:**
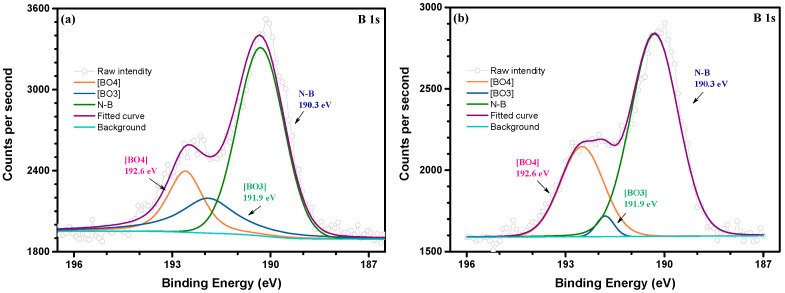
XPS spectrum of B 1 s for (**a**) TC0/CBN composite and (**b**) TC4/CBN composite sintered at 950 °C for 1 h.

**Figure 5 micromachines-14-00303-f005:**
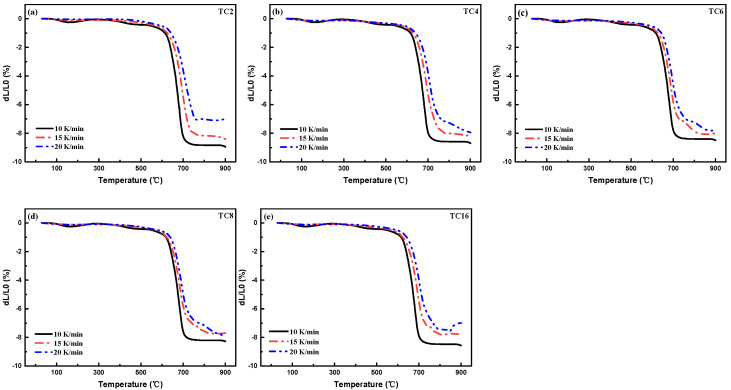
Shrinkage curves for TTC composite with various TC contents sintered at 950 °C for 1 h, (**a**) TC2, (**b**) TC4, (**c**) TC6, (**d**) TC8, and (**e**) TC16.

**Figure 6 micromachines-14-00303-f006:**
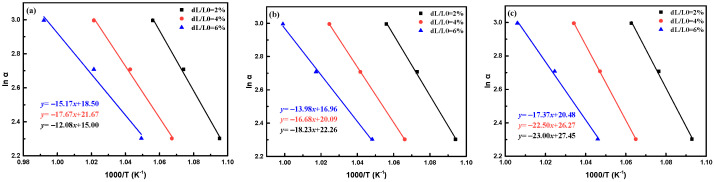
Fitted results of ln α~1000/T plots and the sintering activation energy of the TC composites with various TC contents (**a**) TC2, (**b**) TC4, (**c**) TC6, (**d**) TC8, (**e**) TC16, and (**f**) the results of sintering activation energy of the TC composites..

**Figure 7 micromachines-14-00303-f007:**
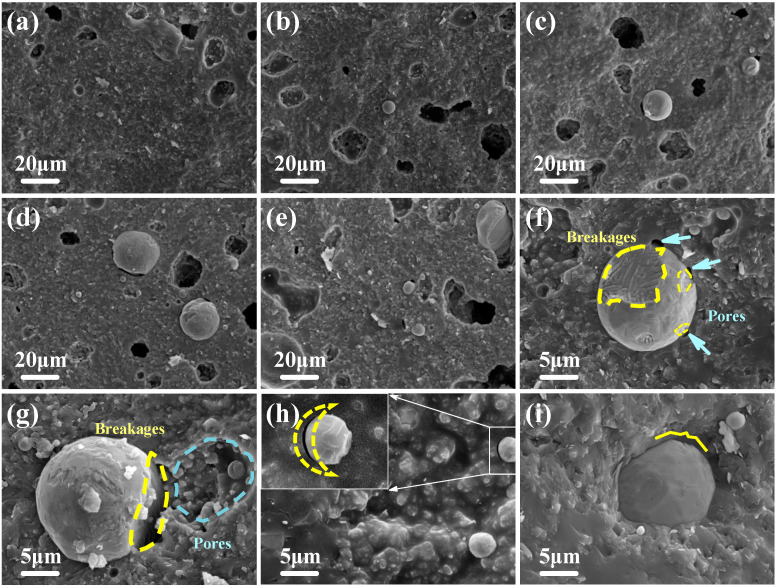
Microstructure and porosification behavior of TC composites, (**a**) TC2, (**b**) TC4, (**c**) TC6, (**d**) TC8, (**e**) TC16, (**f**) alloy powder with breakages, (**g**) voids accompanied by the alloy breakage, (**h**) crescent-shaped gap, and (**i**) dense interface.

**Figure 8 micromachines-14-00303-f008:**
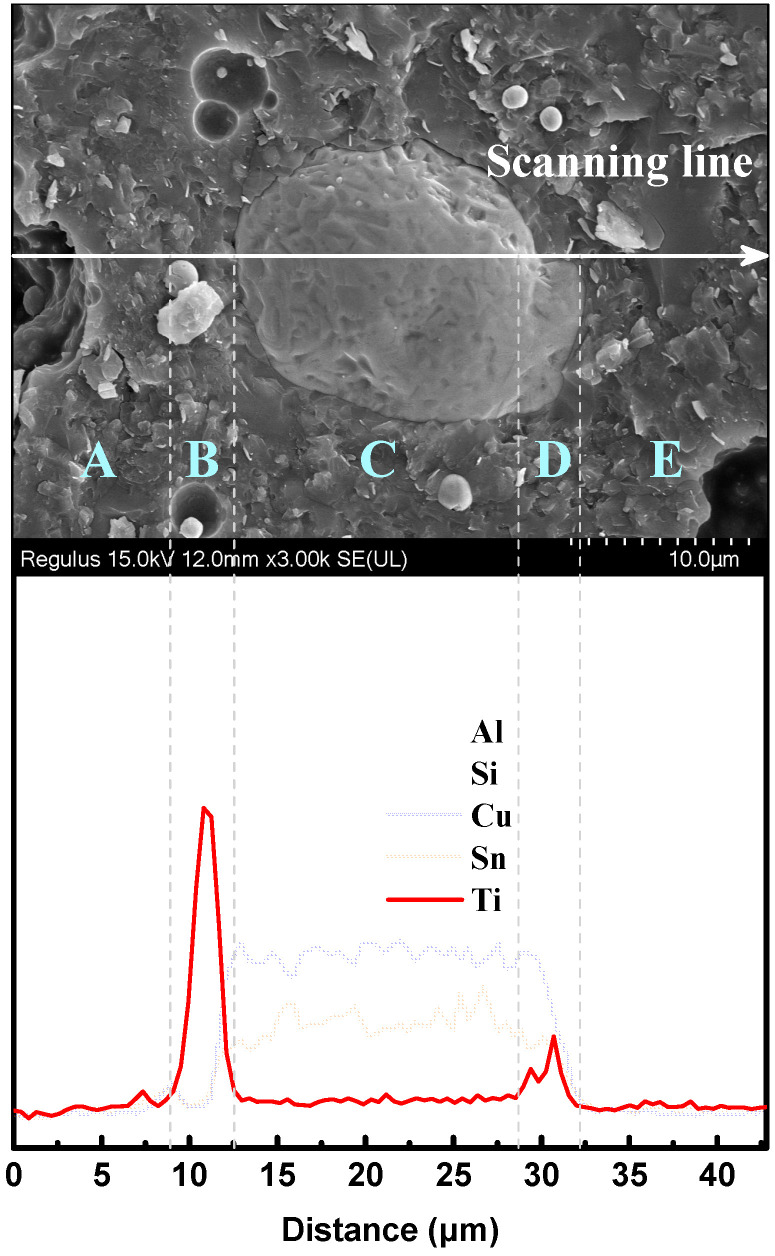
Microstructure and EDS result of interface between the TC particle and the matrix.

**Figure 9 micromachines-14-00303-f009:**
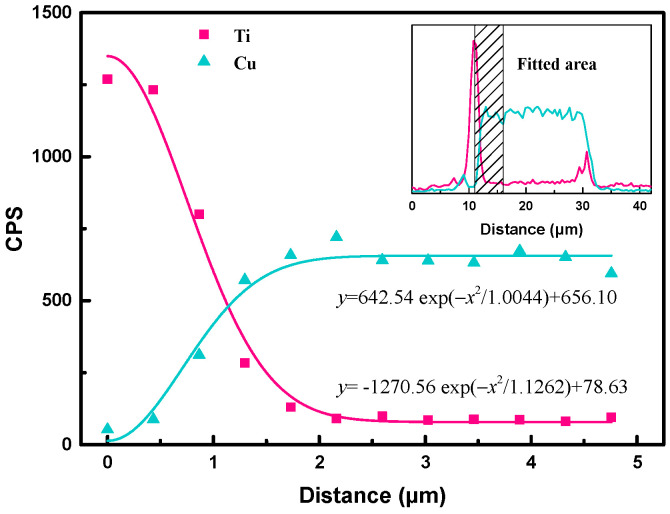
Data points of Ti and Cu from line scanning EDS analysis and their fitted curves.

**Figure 10 micromachines-14-00303-f010:**
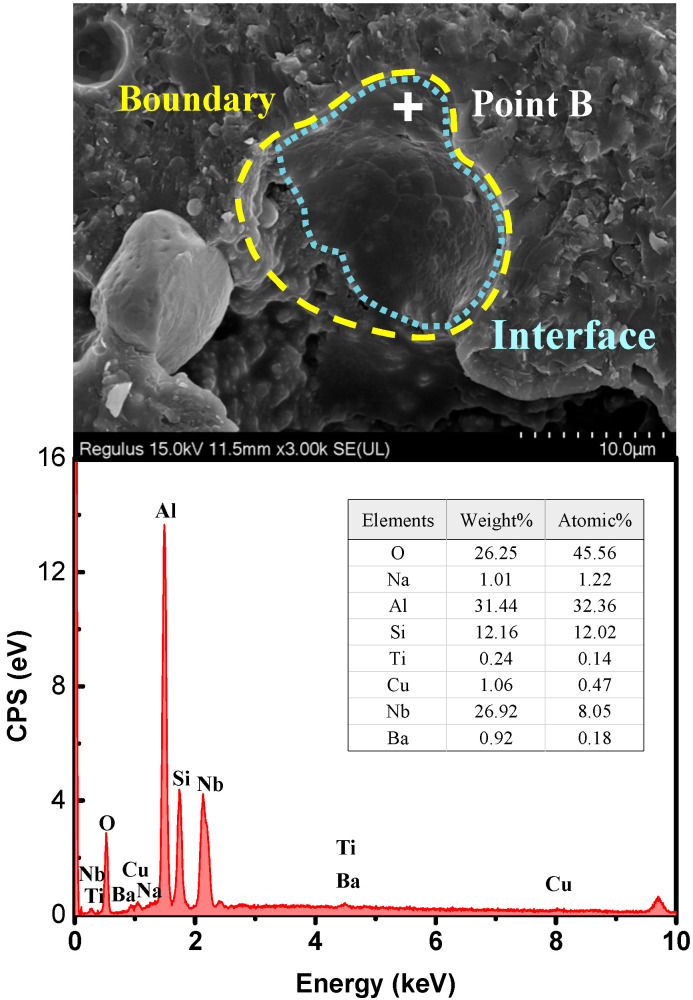
Microstructure and EDS result of the crater left by the TC particle.

**Figure 11 micromachines-14-00303-f011:**
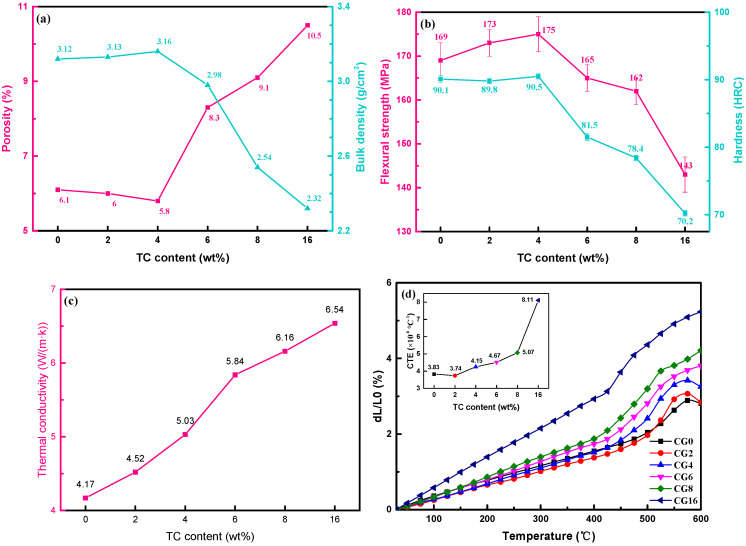
Physical properties of TC composites as a function of TC alloy mass fractions sintered at 950 °C for 1 h: (**a**) porosity and bulk density, (**b**) flexural strength and Rockwell hardness, (**c**) thermal conductivity and (**d**) CTE (30 to 300 °C).

**Figure 12 micromachines-14-00303-f012:**
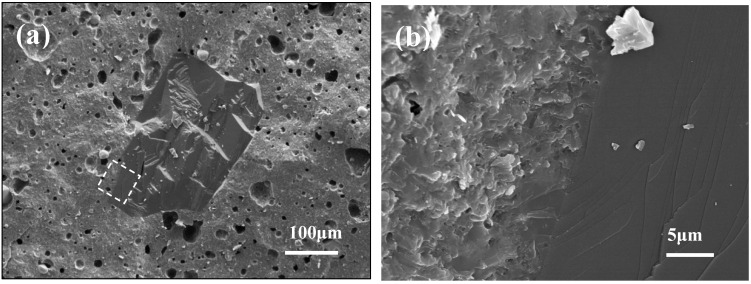
Microstructure of the TC2/CBN composite after flexural strength testing, (**a**) low magnification and (**b**) high magnification.

**Figure 13 micromachines-14-00303-f013:**
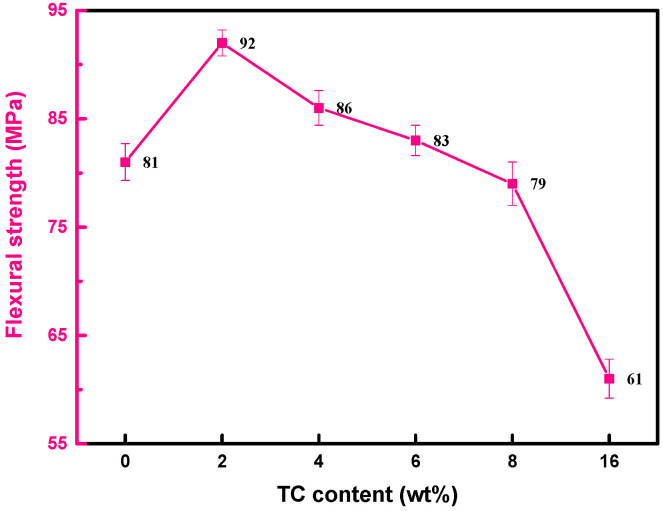
Flexural strength of the TC composites with various TC contents sintered at 950 °C for 1 h.

**Figure 14 micromachines-14-00303-f014:**
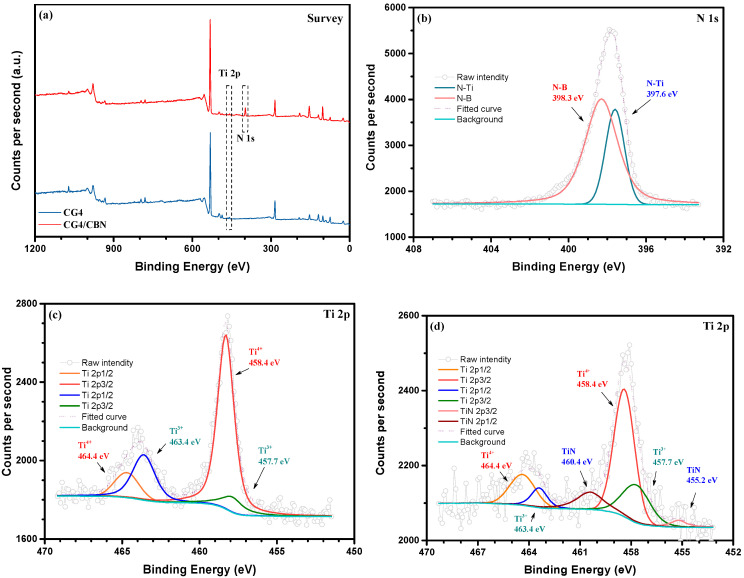
XPS spectrum of the TC composite. (**a**) survey of TC4 and TC4/CBN composite, (**b**) N 1 s of TC4/CBN composite, (**c**) Ti 2p of the matrix added 4 wt% TC alloy, (**d**) Ti 2p of TC4/CBN composite.

**Table 1 micromachines-14-00303-t001:** Compositions of the TC composites (wt%).

Sample	Ti Containing Cu-Based Alloy	Ceramic Filler	Vitrified Bond							
		Al_2_O_3_	SiO_2_	Al_2_O_3_	B_2_O_3_	BaO	Na_2_O	Li_2_O	ZnO	MgO
			45	8	20	8	8	6	3	2
		57.5	42.5							
TC0	0	100								
TC2	2	98								
TC4	4	96								
TC6	6	94								
TC8	8	92								
TC16	16	84								

**Table 2 micromachines-14-00303-t002:** Phase content and relative crystallinity calculated from XRD.

Sample	Crystal Phase Composition (wt%)					Relative Crystallinity (%)
	BaAl_2_Si_2_O_8_	Li_2_Al_2_Si_3_O_10_	Al_2_O_3_	TiO_2_	Sn_11_Cu_39_	
TC2	48.8	14.4	35.3	0.3	1.2	68.2
TC4	48.6	15.2	32.7	0.8	2.7	69.7
TC6	49.3	17.4	28.4	1.7	3.2	71.5
TC8	47.7	19.8	26.7	2.0	3.8	73.2
TC16	32.8	26.4	25.9	3.9	11.0	75.5

## Data Availability

No Statement.

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
