# Peer review of "Effects of Ti Containing Cu-Based Alloy on Sintering Mechanism, Element Diffusion Behavior and Physical Properties of Glass-Ceramic Bond for Cubic Boron Nitride Abrasive Tool Materials"

_micromachines, 2023, doi:10.3390/mi14020303_

Round 1

Reviewer 1 Report

1. In the title: the word (properties) is a general word, it is better if you can mention which properties you need to show

2. In the abstract: By establishing mathematical model.....can you mention the name of the model (if available)

3. In the introduction, can you show the importance of this work

4. section 2.1: can you show the purity of the oxdies

5. In section 2, I found only one equations, can you add the equations used in the calculations of the different parameters 

6. please check the subscript in the text (see for example [SiO4)

7. please improve the resolution of Fig.4

8. The conclusion is long, you can reduce it

9. Please update some References and add some recent Refs (2020-2023)

Reviewer 2 Report

The authors presented an article about “Effects of Ti containing Cu-based alloy on sintering behavior and properties of glass-ceramic bond for cubic boron nitride abrasive tool materials”. The authors have done a very successful article study. Congratulations for that. I saw that it was prepared carefully and systematically. I would like to convey that I found the issue of XRD PDF numbers to be professional. Figures have been prepared in a very high quality and remarkable way. I think the paper is well organized and is appropriate for “Micromachines” journal but the paper will be ready for publication after minor revision.

·       The abstract looks good. Please include all significance numerical results.

·       In the last paragraph of the introduction, it should be expressed the novelty of the study, the differences from the past in detail.

·       What is the problem? Why was the manuscript written? Please explain the reason in the introduction part.

·       Show the dimensions  and chemical properties of the powders used in a table on the paper.

·       How was the weight percentage of the powders used determined?

·       Improve the conclusion parts. (Support the conclusion part with relevant citations and compare with previous similar studies. Please indicate what highlights your conclusions in the article.)

·       Please fix the typographical and eventual language problems in paper.

·       The paper is well-organized. If your work is convenient for this journal’s context then there are many references from this journal. Cited sources should be primary ones. Namely, indexed area shows the power of a paper and directly your paper’s reliability. Please make regulations in this direction.

*** Authors must consider them properly before submitting the revised manuscript. A point-by-point reply is required when the revised files are submitted.

Reviewer 3 Report

In this work, the authors fabricated Ti containing Cu-based alloy reinforced glass-ceramic bond for cubic boron nitride and studied its crystal composition, phase transformation, sintering activation energy, microstructure, element diffusion mathematical model, physical properties, and the bonding mechanism between the TC alloy reinforced glass-ceramic bond and the CBN grains. The paper is nicely written and introduction contains enough background material to understand the problem statement. The resolution of the figures is very good to understand the results. The finding are supported by the data. Therefore i would like to recommend the paper in its current form. 

Author Response

Thank you very much